# Reshaping Visual Datasets for Domain Adaptation

**Boqing Gong**
U. of Southern California
Los Angeles, CA 90089
boqinggo@usc.edu

**Kristen Grauman**
U. of Texas at Austin
Austin, TX 78701
grauman@cs.utexas.edu

**Fei Sha**
U. of Southern California
Los Angeles, CA 90089
feisha@usc.edu

## Abstract

In visual recognition problems, the common data distribution mismatches between training and testing make domain adaptation essential. However, image data is difficult to manually divide into the discrete domains required by adaptation algorithms, and the standard practice of equating datasets with domains is a weak proxy for all the real conditions that alter the statistics in complex ways (lighting, pose, background, resolution, etc.) We propose an approach to automatically discover latent domains in image or video datasets. Our formulation imposes two key properties on domains: *maximum distinctiveness* and *maximum learnability*. By maximum distinctiveness, we require the underlying distributions of the identified domains to be different from each other to the maximum extent; by maximum learnability, we ensure that a strong discriminative model can be learned from the domain. We devise a nonparametric formulation and efficient optimization procedure that can successfully discover domains among both training and test data. We extensively evaluate our approach on object recognition and human activity recognition tasks.

## 1   Introduction

A *domain* refers to an underlying data distribution. Generally, there are two: the one with which classifiers are trained, and the other to which classifiers are applied. While many learning algorithms assume the two are the same, in real-world applications, the distributions are often mismatched, causing significant performance degradation when the classifiers are applied. Domain adaptation techniques are crucial in building robust classifiers to address mismatched new and unexpected target environments. As such, the subject has been intensively studied in computer vision [1, 2, 3, 4], speech and language processing [5, 6], and statistics and learning [7, 8, 9, 10].

While domain adaptation research largely focuses on how adaptation should proceed, there are also vital questions concerning the domains themselves: *what exactly is a domain composed of?* and *how are domains different from each other*? For some applications, the answers come naturally. For example, in speech recognition, we can organize data into speaker-specific domains where each domain contains a single speaker's utterances. In language processing, we can organize text data into language-specific domains. For those types of data, we can neatly categorize each instance with a discrete set of *semantically meaningful* properties; a domain is thus naturally composed of instances of the same (subset of) properties.

For visual recognition, however, the same is not possible. In addition to large intra-category appearance variations, images and video of objects (or scenes, attributes, activities, etc.) are also significantly affected by many extraneous factors such as pose, illumination, occlusion, camera resolution, and background. Many of these factors simply do not naturally lend themselves to deriving discrete domains. Furthermore, the factors overlap and interact in images in complex ways. In fact, even coming up with a comprehensive set of such properties is a daunting task in its own right—not to mention automatically detecting them in images!

Partially due to these conceptual and practical constraints, datasets for visual recognition are not deliberately collected with clearly identifiable domains [11, 12, 13, 14, 15]. Instead, standard image/video collection is a product of trying to ensure coverage of the target category labels on one hand, and managing resource availability on the other. As a result, a troubling practice in visual domain adaptation research is to equate *datasets* with domains and study the problem of *cross-dataset generalization* or *correcting dataset bias* [16, 17, 18, 19].

One pitfall of this ad hoc practice is that a dataset could be an agglomeration of several distinctive domains. Thus, modeling the dataset as a single domain would necessarily blend the distinctions, potentially damaging visual discrimination. Consider the following human action recognition task, which is also studied empirically in this work. Suppose we have a training set containing videos of multiple subjects taken at view angles of $30°$ and $90°$, respectively. Unaware of the distinction of these two views of videos, a model for the training set as a single training domain needs to account for both inter-subject and inter-view variations. Presumably, applying the model to recognizing videos taken at view angle of $45°$ (*i.e.*, from the test domain) would be less effective than applying models accounting for the two view angles separately, i.e., modeling inter-subject variations only.

How can we avoid such pitfalls? More specifically, *how can we form characteristic domains*, without resorting to the hopeless task of manually defining properties along which to organize them? We propose novel learning methods to automatically reshape datasets into domains. This is a challenging unsupervised learning problem. At the surface, we are not given any information about the domains that the datasets contain, such as the statistical properties of the domains, or even the number of domains. Furthermore, the challenge cannot be construed as a traditional clustering problem; simply clustering images by their appearance is prone to reshaping datasets into per-category domains, as observed in [20] and our own empirical studies. Moreover, there may be many complex factors behind the domains, making it difficult to model the domains with parametric mixture models on which traditional clustering algorithms (e.g., Kmeans or Gaussian mixtures) are based.

Our key insights are two axiomatic properties that latent domains should possess: *maximum distinctiveness* and *maximum learnability*. By *maximum distinctiveness*, we identify domains that are maximally different in distribution from each other. This ensures domains are characteristic in terms of their large inter-domain variations. By *maximum learnability*, we identify domains from which we can derive strong discriminative models to apply to new testing data.

In section 2, we describe our learning methods for extracting domains with these desirable properties. We derive nonparametric approaches to measure domain discrepancies and show how to optimize them to arrive at maximum distinctiveness. We also show how to achieve maximum learnability by monitoring an extracted domain's discriminative learning performance, and we use that property to automatically choose the number of latent domains. To our best knowledge, [20] is the first and only work addressing latent domain discovery. We postpone a detailed discussion and comparison to their method to section 3, after we have described our own.

In section 4, we demonstrate the effectiveness of our approach on several domain adaptation tasks for object recognition and human activity recognition. We show that we achieve far better classification results using adapted classifiers learned on the discovered domains. We conclude in section 5.

## 2  Proposed approach

We assume that we have access to one or more annotated datasets with a total of $\mathsf{M}$ data instances. The data instances are in the form of $(\boldsymbol{x}_m, y_m)$ where $\boldsymbol{x}_m \in \mathbb{R}^{\mathsf{D}}$ is the feature vector and $y_m \in [\mathsf{C}]$ the corresponding label out of $\mathsf{C}$ categories. Moreover, we assume that each data instance comes from a latent domain $z_m \in [\mathsf{K}]$ where $\mathsf{K}$ is the number of domains.

In what follows, we start by describing our algorithm for inferring $z_m$ assuming $\mathsf{K}$ is known. Then we describe how to infer $\mathsf{K}$ from the data.

### 2.1  Maximally distinctive domains

Given $\mathsf{K}$, we denote the distributions of unknown domains $\mathcal{D}_k$ by $P_k(\boldsymbol{x}, y)$ for $k \in [\mathsf{K}]$. We do not impose any parametric form on $P_k(\cdot, \cdot)$. Instead, the marginal distribution $P_k(\boldsymbol{x})$ is approximated

by the empirical distribution $\hat{P}_k(\boldsymbol{x})$

$$\hat{P}_k(\boldsymbol{x}) = \frac{1}{\mathsf{M}_k} \sum_m \delta_{\boldsymbol{x}_m} z_{mk},$$

where $\mathsf{M}_k$ is the number of data instances to be assigned to the domain $k$ and $\delta_{\boldsymbol{x}_m}$ is an atom at $\boldsymbol{x}_m$. $z_{mk} \in \{0, 1\}$ is a binary indicator variable and takes the value of 1 when $z_m = k$. Note that $\mathsf{M}_k = \sum_m z_{mk}$ and $\sum_k \mathsf{M}_k = \mathsf{M}$.

What kind of properties do we expect from $\hat{P}_k(\boldsymbol{x})$? Intuitively, we would like any two different domains $\hat{P}_k(\boldsymbol{x})$ and $\hat{P}_{k'}(\boldsymbol{x})$ to be as distinctive as possible. In the context of modeling visual data, this implies that intra-class variations between domains are often far more pronounced than inter-class variations within the same domain. As a concrete example, consider the task of differentiating commercial jetliners from fighter jets. While the two categories are easily distinguishable when viewed from the same pose (frontal view, side view, etc.), there is a significant change in appearance when either category undergoes a pose change. Clearly, defining domains by simply clustering the images by appearance is insufficient; the inter-category and inter-pose variations will both contribute to the clustering procedure and may lead to unreasonable clusters. Instead, to identify characteristic domains, we need to look for divisions of the data that yield ***maximally distinctive*** distributions.

To quantify this intuition, we need a way to measure the difference in distributions. To this end, we apply a kernel-based method to examine whether two samples are from the same distribution [21]. Concretely, let $k(\cdot, \cdot)$ denote a characteristic positive semidefinite kernel (such as the Gaussian kernel). We compute the the difference between the means of two empirical distributions in the reproducing kernel Hilbert space (RKHS) $\mathcal{H}$ induced by the kernel function,

$$d(k, k') = \left\| \frac{1}{\mathsf{M}_k} \sum_m k(\cdot, \boldsymbol{x}_m) z_{mk} - \frac{1}{\mathsf{M}'_k} \sum_m k(\cdot, \boldsymbol{x}_m) z_{mk'} \right\|_{\mathcal{H}}^2 \tag{1}$$

where $k(\cdot, \boldsymbol{x}_m)$ is the image (or kernel-induced feature) of $\boldsymbol{x}_m$ under the kernel. The measure approaches zero as the number of samples tends to infinity, if and only if the two domains are the same, $P_k = P_{k'}$. We define the *total domain distinctiveness* (TDD) as the sum of this quantity over all possible pairs of domains:

$$\mathsf{TDD}(\mathsf{K}) = \sum_{k \neq k'} d(k, k'), \tag{2}$$

and choose domain assignments for $z_m$ such that TDD is maximized. We first discuss this optimization problem in its native formulation of integer programming, followed by a more computationally convenient continuous optimization.

**Optimization** In addition to the binary constraints on $z_{mk}$, we also enforce

$$\sum_{k=1}^{\mathsf{K}} z_{mk} = 1, \quad \forall\, m \in [\mathsf{M}], \text{ and } \frac{1}{\mathsf{M}_k} \sum_{m=1}^{\mathsf{M}} z_{mk} y_{mc} = \frac{1}{\mathsf{M}} \sum_{m=1}^{\mathsf{M}} y_{mc}, \quad \forall\, c \in [\mathsf{C}], \quad k \in [\mathsf{K}] \tag{3}$$

where $y_{mc}$ is a binary indicator variable, taking the value of 1 if $y_m = c$.

The first constraint stipulates that every instance will be assigned to one domain and one domain only. The second constraint, which we refer to as the *label prior constraint* (LPC), requires that within each domain, the class labels are distributed according to the prior distribution (of the labels), estimated empirically from the labeled data.

LPC does not restrict the absolute numbers of instances of different labels in each domain. It only reflects the intuition that in the process of data collection, the relative percentages of different classes are approximately in accordance with a prior distribution that is independent of domains. For example, in action recognition, if the "walking" category occurs relatively frequently in a domain corresponding to brightly lit video, we also expect it to be frequent in the darker videos. Thus, when data instances are re-arranged into latent domains, the same percentages are likely to be preserved.

The optimization problem is NP-hard due to the integer constraints. In the following, we relax it into a continuous optimization, which is more accessible with off-the-shelf optimization packages.

**Relaxation**    We introduce new variables $\beta_{mk} = z_{mk}/\mathsf{M}_k$, and relax them to live on the simplex

$$\boldsymbol{\beta}_k = (\beta_{1k}, \cdots, \beta_{\mathsf{M}k})^{\mathrm{T}} \in \Delta = \left\{ \boldsymbol{\beta}_k : \beta_{mk} \geq 0, \sum_{m=1}^{\mathsf{M}} \beta_{mk} = 1 \right\}$$

for $k = 1, \cdots, \mathsf{K}$. With the new variables, our optimization problem becomes

$$\max_{\boldsymbol{\beta}} \quad \sum_{k \neq k'} \mathsf{TDD}(\mathsf{K}) = \sum_{k \neq k'} (\boldsymbol{\beta}_k - \boldsymbol{\beta}_{k'})^{\mathrm{T}} \boldsymbol{K} (\boldsymbol{\beta}_k - \boldsymbol{\beta}_{k'}) \tag{4}$$

$$\text{s.t.} \quad 1/\mathsf{M} \leq \sum_k \beta_{mk} \leq 1/\mathsf{C}, \quad m = 1, 2, \cdots, \mathsf{M}, \tag{5}$$

$$(1-\delta)/\mathsf{M} \sum_m y_{mc} \leq \sum_m \beta_{mk} y_{mc} \leq (1+\delta)/\mathsf{M} \sum_m y_{mc}, \quad c = 1, \cdots, \mathsf{C}, \quad k = 1, \cdots, \mathsf{K},$$

where $\boldsymbol{K}$ is the $\mathsf{M} \times \mathsf{M}$ kernel matrix. The first constraint stems from the (default) requirement that every domain should have at least one instance per category, namely, $\mathsf{M}_k \geq \mathsf{C}$ and every domain should at most have M instances ($\mathsf{M}_k \leq \mathsf{M}$). The second constraint is a relaxed version of the LPC, allowing a small deviation from the prior distribution by setting $\delta = 1\%$. We assign $\boldsymbol{x}_m$ to the domain $k$ for which $\beta_{mk}$ is the maximum of $\beta_{m1}, \cdots, \beta_{m\mathsf{K}}$.

This relaxed optimization problem is a maximization of convex quadratic function subject to linear constraints. Though in general still NP-hard, this type of optimization problem has been studied extensively and we have found existing solvers are adequate in yielding satisfactory solutions.

## 2.2    Maximally learnable domains: determining the number of domains

Given M instances, how many domains hide inside? Note that the total domain distinctiveness $\mathsf{TDD}(\mathsf{K})$ increases as K increases — presumably, in the extreme case, each domain has only a few instances and their distributions would be maximally different from each other. However, such tiny domains would offer insufficient data to separate the categories of interest reliably.

To infer the optimal K, we appeal to *maximum learnability*, another desirable property we impose on the identified domains. Specifically, for any identified domain, we would like the data instances it contains to be adequate to build a strong classifier for labeled data — failing to do so would cripple the domain's adaptability to new test data.

Following this line of reasoning, we propose domain-wise cross-validation (DWCV) to identify the optimal K. DWCV consists of the following steps. First, starting from $\mathsf{K} = 2$, we use the method described in the previous section to identify K domains. Second, for each identified domain, we build discriminative classifiers, using the label information and evaluate them with cross-validation. Denote the cross-validation accuracy for the $k$-th domain by $A_k$. We then combine all the accuracies with a weighted sum

$$A(\mathsf{K}) = 1/\mathsf{M} \sum_{k=1}^{\mathsf{K}} \mathsf{M}_k A_k.$$

For very large K such that each domain contains only a few examples, $A(\mathsf{K})$ approaches the classification accuracy using the class prior probability to classify. Thus, starting at $\mathsf{K} = 2$ (and assuming $A(2)$ is greater than the prior probability's classification accuracy), we choose $\mathsf{K}^*$ as the value that attains the highest cross-validation accuracy: $\mathsf{K}^* = \arg\max_{\mathsf{K}} A(\mathsf{K})$. For N-fold cross-validation, a practical bound for the largest K we need to examine is $\mathsf{K}_{\max} \leq \min\{\mathsf{M}/(\mathsf{N}\mathsf{C}), \mathsf{C}\}$. Beyond this bound it does not quite make sense to do cross-validation.

## 3    Related work

Domain adaptation is a fundamental research subject in statistical machine learning [9, 22, 23, 10], and is also extensively studied in speech and language processing [5, 6, 8] and computer vision [1, 2, 3, 4, 24, 25]. Mostly these approaches are validated by adapting between datasets, which, as discussed above, do not necessarily correspond to well-defined domains.

In our previous work, we proposed to identify some landmark data points in the source domain which are distributed similarly to the target domain [26]. While that approach also slices the training set, it differs in the objective. We discover the underlying domains of the training datasets, each of which will be adaptable, whereas the landmarks in [26] are intentionally biased towards the single given target domain. Hoffman *et al.*'s work [20] is the most relevant to ours. They also aim at discovering the latent domains from datasets, by modeling the data with a hierarchical distribution consisting of Gaussian mixtures. However, their explicit form of distribution may not be easily satisfiable in real data. In contrast, we appeal to nonparametric methods, overcoming this limitation without assuming any form of distribution. In addition, we examine the new scenario where the test set is also composed of heterogeneous domains.

A generalized clustering approach by Jegelka *et al.* [27] shares the idea of maximum distinctiveness (or "discriminability" used in [27]) criterion with our approach. However, their focus is the setting of unsupervised clustering where ours is domain discovery. As such, they adopt a different regularization term from ours, which exploits labels in the datasets.

Multi-domain adaptation methods suppose that multiple source domains are given as input, and the learner must adapt from (some of) them to do well in testing on a novel target domain [28, 29, 10]. In contrast, in the problem we tackle, the division of data into domains is not given—our algorithm must discover the latent domains. After our approach slices the training data into multiple domains, it is natural to apply multi-domain techniques to achieve good performance on a test domain. We will present some related experiments in the next section.

## 4 Experimental Results

We validate our approach on visual object recognition and human activity recognition tasks. We first describe our experimental settings, and then report the results of identifying latent domains and using the identified domains for adapting classifiers to a new mono-domain test set. After that, we present and report experimental results of reshaping heterogeneous test datasets into domains matching to the identified training domains. Finally, we give some qualitative analyses and details on choosing the number of domains.

### 4.1 Experimental setting

**Data** For object recognition, we use images from Caltech-256 (C) [14] and the image datasets of Amazon (A), DSLR (D), and Webcam (W) provided by Saenko *et al.* [2]. There are total 10 common categories among the 4 datasets. These images mainly differ in the data collection sources: Caltech-256 was collected from webpages on the Internet, Amazon images from amazon.com, and DSLR and Webcam images from an office environment. We represent images with bag-of-visual-words descriptors following previous work on domain adaptation [2, 4]. In particular, we extract SURF [30] features from the images, use K-means to build a codebook of 800 clusters, and finally obtain an 800-bin histogram for each image.

For action recognition from video sequences, we use the IXMAS multi-view action dataset [15]. There are five views (Camera $0, 1, \cdots, 4$) of eleven actions in the dataset. Each action is performed three times by twelve actors and is captured by the five cameras. We keep the first five actions performed by *alba, andreas, daniel, hedlena, julien*, and *nicolas* such that the irregularly performed actions [15] are excluded. In each view, 20 sequences are randomly selected per actor per action. We use the shape-flow descriptors to characterize the motion of the actions [31].

**Evaluation strategy** The four image datasets are commonly used as distinctive domains in research in visual domain adaptation [2, 3, 4, 32]. Likewise, each view in the IXMAS dataset is often taken as a domain in action recognition [33, 34, 35, 24]. Similarly, in our experiments, we use a subset of these datasets (views) as source domains for training classifiers and the rest of the datasets (views) as target domains for testing. However, the key difference is that we do not compare performance of different adaptation algorithms which assume domains are already given. Instead, we evaluate the effectiveness of our approach by investigating whether its automatically identified domains improve adaptation, that is, whether recognition accuracy on the target domains can be improved by reshaping the datasets into their latent source domains.

Table 1: Oracle recognition accuracy on target domains by adapting original or identified domains

| $\mathcal{S}$ | A, C | D, W | C, D, W | Cam 0, 1 | Cam 2, 3, 4 |
|---|---|---|---|---|---|
| $\mathcal{T}$ | D, W | A, C | A | Cam 2, 3, 4 | Cam 0, 1 |
| $G_{\text{ORIG}}$ | 41.0 | 32.6 | 41.8 | 44.6 | 47.1 |
| $G_{\text{OTHER}}$ [20] | 39.5 | 33.7 | 34.6 | 43.9 | 45.1 |
| $G_{\text{OURS}}$ | 42.6 | 35.5 | 44.6 | 47.3 | 50.3 |

Table 2: Adaptation recognition accuracies, using original and identified domains with different multi-source adaptation methods

| Latent Domains | Multi-DA method | A, C | D, W | C, D, W | Cam 0, 1 | Cam 2, 3, 4 |
|---|---|---|---|---|---|---|
| | | D, W | A, C | A | Cam 2, 3, 4 | Cam 0, 1 |
| ORIGINAL | UNION | 41.7 | 35.8 | 41.0 | 45.1 | 47.8 |
| [20] | ENSEMBLE | 31.7 | 34.4 | 38.9 | 43.3 | 29.6 |
| | MATCHING | 39.6 | 34.0 | 34.6 | 43.2 | 45.2 |
| OURS | ENSEMBLE | 38.7 | 35.8 | 42.8 | 45.0 | 40.5 |
| | MATCHING | 42.6 | 35.5 | 44.6 | 47.3 | 50.3 |

We use the geodesic flow kernel for adapting classifiers [4]. To use the kernel-based method for computing distribution difference, we use Gaussian kernels, cf. section 2. We set the kernel bandwidth to be twice the median distances of all pairwise data points. The number of latent domains K is determined by the DWCV procedure (cf. section 2.2).

## 4.2 Identifying latent domains from training datasets

**Notation** Let $\mathcal{S} = \{\mathcal{S}_1, \mathcal{S}_2, \ldots, \mathcal{S}_J\}$ denote the J datasets we will be using as training source datasets and let $\mathcal{T} = \{\mathcal{T}_1, \mathcal{T}_2, \ldots, \mathcal{T}_L\}$ denote the L datasets we will be using as testing target datasets. Furthermore, let K denote the number of optimal domains discovered by our DWCV procedure and $\mathcal{U} = \{\mathcal{U}_1, \mathcal{U}_2, \ldots, \mathcal{U}_K\}$ the K hidden domains identified by our approach. Let $r(\mathcal{A} \to \mathcal{B})$ denote the recognition accuracy on the target domain $\mathcal{B}$ with $\mathcal{A}$ as the source domain.

**Goodness of the identified domains** We examine whether $\{\mathcal{U}_k\}$ is a set of good domains by computing the expected best possible accuracy of using the identified domains *separately* for adaptation

$$G_{\text{OURS}} = \mathbb{E}_{\mathcal{B} \in \mathcal{P}} \max_k r(\mathcal{U}_k, \mathcal{B}) \approx \frac{1}{\mathsf{L}} \sum_l \max_k r(\mathcal{U}_k \to \mathcal{T}_l) \qquad (6)$$

where $\mathcal{B}$ is a target domain drawn from a distribution on domains $\mathcal{P}$. Since this distribution is not obtainable, we approximate the expectation with the empirical average over the observed testing datasets $\{\mathcal{T}_l\}$. Likewise, we can define $G_{\text{ORIG}}$ where we compute the best possible accuracy for the original domains $\{\mathcal{S}_j\}$, and $G_{\text{OTHER}}$ where we compute the same quantity for a competing method for identifying latent domains, proposed in [20]. Note that the $\max$ operation requires that the target domains be annotated; thus the accuracies are the most optimistic estimate for all methods, and upper bounds of practical algorithms.

Table 1 reports the three quantities on different pairs of sources and target domains. Clearly, our method yields a better set of identified domains, which are always better than the original datasets. We also experimented using Kmeans or random partition for clustering data instances into domains. Neither yields competitive performance and the results are omitted here for brevity.

**Practical utility of identified domains** In practical applications of domain adaptation algorithms, however, the target domains are not annotated. The oracle accuracies reported in Table 1 are thus not achievable in general. In the following, we examine how closely the performance of the identified domains can approximate the oracle if we employ multi-source adaptation.

To this end, we consider several choices of multiple-source domain adaptation methods:

- UNION The most naive way is to combine all the source domains into a single dataset and adapt from this "mega" domain to the target domains. We use this as a baseline.
- ENSEMBLE A more sophisticated strategy is to adapt each source domain to the target domain and combine the adaptation results in the form of combining multiple classifiers [20].

Table 3: Results of reshaping the test set when it consists of data from multiple domains.

| | From identified (Reshaping training only) | | | No reshaping | Conditional reshaping |
|---|---|---|---|---|---|
| | $A' \to F$ | $B' \to F$ | $C' \to F$ | $A \bigcup B \bigcup C \to F$ | $X \to F_X, \forall X \in \{A', B', C'\}$ |
| Cam 012 | 36.4 | 37.1 | 37.7 | 37.3 | 38.5 |
| Cam 123 | 40.4 | 38.7 | 39.6 | 39.9 | 41.1 |
| Cam 234 | 46.5 | 45.7 | 46.1 | 47.8 | 49.2 |
| Cam 340 | 50.7 | 50.6 | 50.5 | 52.3 | 54.9 |
| Cam 401 | 43.6 | 41.8 | 43.9 | 43.3 | 44.8 |

- MATCHING This strategy compares the empirical (marginal) distribution of the source domains and the target domains and selects the *single source domain* that has the smallest difference to the target domain to adapt. We use the kernel-based method to compare distributions, as explained in section 2. Note that since we compare only the marginal distributions, we do not require the target domains to be annotated.

Table 2 reports the averaged recognition accuracies on the target domains, using either the original datasets/domains or the identified domains as the source domains. The latent domains identified by our method generally perform well, especially using MATCHING to select the single best source domain to match the target domain for adaptation. In fact, contrasting Table 2 to Table 1, the MATCHING strategy for adaptation is able to match the oracle accuracies, even though the matching process does not use label information from the target domains.

## 4.3 Reshaping the test datasets

So far we have been concentrating on reshaping multiple annotated datasets (for training classifiers) into domains for adapting to test datasets. However, test datasets can also be made of multiple latent domains. Hence, it is also instrumental to investigate whether we can reshape the test datasets into multiple domains to achieve better adaptation results.

However, the reshaping process for test datasets has a critical difference from reshaping training datasets. Specifically, we should reshape test datasets, *conditioning* on the identified domains from the training datasets — the goal is to discover latent domains in the test datasets that match the domains in the training datasets as much as possible. We term this *conditional reshaping*.

Computationally, conditional reshaping is more tractable than identifying latent domains from the training datasets. Concretely, we minimize the distribution differences between the latent domains in the test datasets and the domains in the training datasets, using the kernel-based measure explained in section 2. The optimization problem, however, can be relaxed into a convex quadratic programming problem. Details are in the Suppl. Material.

Table 3 demonstrates the benefit of conditionally reshaping the test datasets, on cross-view action recognition. This problem inherently needs test set reshaping, since the person may be viewed from any direction at test time. (In contrast, test sets for the object recognition datasets above are less heterogeneous.) The first column shows five groups of training datasets, each being a different view, denoted by $A, B$ and $C$. In each group, the remaining views $D$ and $E$ are merged into a new test dataset, denoted by $F = D \bigcup E$.

Two baselines are included: (1) adapting from the identified domains $A', B'$ and $C'$ to the merged dataset $F$; (2) adapting from the merged dataset $A \bigcup B \bigcup C$ to $F$. These are contrasted to adapting from the identified domains in the training datasets to the *matched* domains in $F$. In most groups, there is a significant improvement in recognition accuracies by conditional reshaping over no reshaping on either training or testing, and reshaping on training only.

## 4.4 Analysis of identified domains and the optimal number of domains

It is also interesting to see which factors are dominant in the identified domains. Object appearance, illumination, or background? Do they coincide with the factors controlled by the dataset collectors?

Some exemplar images are shown in Figure 1, where each row corresponds to an original dataset, and each column is an identified domain across two datasets. On the left of Figure 1 we reshape Amazon and Caltech-256 into two domains. In Domain II all the "laptop" images 1) are taken from

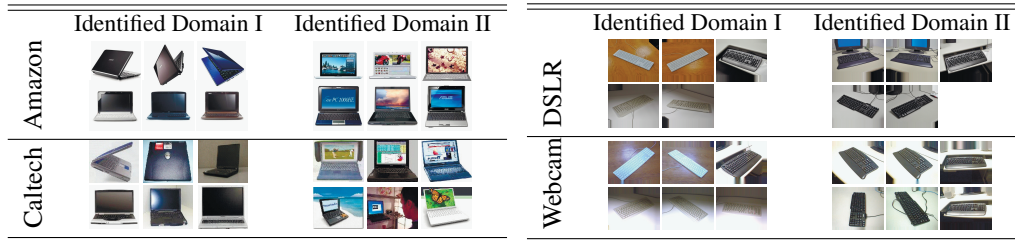

Figure 1: Exemplar images from the original and identified domains after reshaping. Note that identified domains contain images from both datasets.

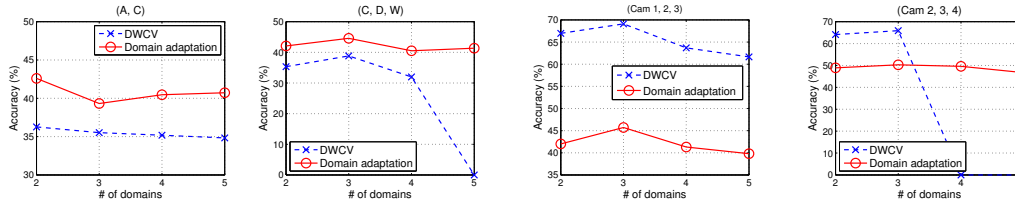

Figure 2: Domain-wise cross-validation (DWCV) for choosing the number of domains.

the front view and 2) have colorful screens, while Domain I images are less colorful and have more diversified views. It looks like the domains in Amazon and Caltech-256 are mainly determined by the factors of *object pose* and *appearance (color)*.

The figures on the right are from reshaping DSLR and Webcam, of which the "keyboard" images are taken in an office environment with various lighting, object poses, and background controlled by the dataset creators [2]. We can see that the images in Domain II have gray background, while in Domain I the background is either white or wooden. Besides, keyboards of the same model, characterized by color and shape, are almost perfectly assigned to the same domain. In sum, the main factors here are probably *background* and *object appearance (color and shape)*.

Figure 2 plots some intermediate results of the domain-wise cross-validation (DWCV) for determining the number of domains K to identify from the multiple training datasets. In addition to the DWCV accuracy $A(K)$, the average classification accuracies on the target domain(s) are also included for reference. We set $A(K)$ to 0 when some categories in a domain are assigned with only one or no data point (as a result of optimization). Generally, $A(K)$ goes up and then drops at some point, before which is the optimal $K^\star$ we use in the experiments. Interestingly, the number favored by DWCV coincides with the number of datasets we mix, even though, as our experiments above show, the ideal domain boundaries do not coincide with the dataset boundaries.

## 5  Conclusion

We introduced two domain properties, *maximum distinctiveness* and *maximum learnability*, to discover latent domains from datasets. Accordingly, we proposed nonparametric approaches encouraging the extracted domains to satisfy these properties. Since in each domain visual discrimination is more consistent than that in the heterogeneous datasets, better prediction performance can be achieved on the target domain. The proposed approach is extensively evaluated on visual object recognition and human activity recognition tasks. Our identified domains outperform not only the original datasets but also the domains discovered by [20], validating the effectiveness of our approach. It may also shed light on dataset construction in the future by examining the main factors of the domains discovered from the existing datasets.

**Acknowledgments** K.G is supported by ONR ATL N00014-11-1-0105. B.G. and F.S. is supported by ARO Award# W911NF-12-1-0241 and DARPA Contract# D11AP00278 and the IARPA via DoD/ARL contract # W911NF-12-C-0012. The U.S. Government is authorized to reproduce and distribute reprints for Governmental purposes notwithstanding any copyright annotation thereon. The views and conclusions contained herein are those of the authors and should not be interpreted as necessarily representing the official policies or endorsements, either expressed or implied, of IARPA, DoD/ARL, or the U.S. Government.

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
