[Supplementary Material · supp.pdf]

# Supplementary Material for
# Reshaping Visual Datasets for Domain Adaptation

**Boqing Gong**
Dept. of Computer Science
U. of Southern California
Los Angeles, CA 90089
boqinggo@usc.edu

**Kristen Grauman**
Dept. of Computer Science
U. of Texas at Austin
Austin, TX 78701
grauman@cs.utexas.edu

**Fei Sha**
Dept. of Computer Science
U. of Southern California
Los Angeles, CA 90089
feisha@usc.edu

## Reshaping the test datasets

When the test datasets are made of multiple latent domains, adaptation by blindly regarding them as a single domain might only achieve suboptimal performance, since this would blend the visual discriminations in the latent domains. To overcome this issue, we propose to reshape the test datasets too into multiple domains.

However, the reshaping process for test datasets has a critical difference from reshaping training datasets. Specifically, we should reshape test datasets, *conditioning* on the identified domains from the training datasets — the goal is to discover latent domains in the test datasets that match the domains in the training datasets as much as possible. We term this ***conditional reshaping***.

Concretely, given the test data $\{(\boldsymbol{x}_n^t, y_n^t)\}_{n=1}^{\mathsf{N}}$ potentially drawn from multiple domains, we introduce indicator variables $z_{nv}^t \in \{0, 1\}$, $n = 1, \cdots, \mathsf{N}$, $v = 1, \cdots, \mathsf{K}$ for the test data points, and solve for the variables by minimizing the empirical distance between the distribution embeddings of an identified training domain and an latent domain of the test datasets, respectively,

$$\min_{\{z_{nv}^t\}} \quad \sum_{v=1}^{\mathsf{K}} \left\| \frac{1}{\mathsf{M}_v} \sum_{m=1}^{\mathsf{M}} z_{mv}\phi(\boldsymbol{x}_m) - \frac{1}{\sum_{j=1}^{\mathsf{N}} z_{jv}^t} \sum_{n=1}^{\mathsf{N}} z_{nv}^t\phi(\boldsymbol{x}_n^t) \right\|_{\mathcal{H}}^2, \tag{1}$$

where the values of $z_{mv}$'s are inherited from Section 2 in the main text. Note that we do not impose any balance constraints on $z_{nv}^t$, allowing the number of test domains smaller than that of the training domains.

With $\beta_{nv}^t = z_{nv}^t / \sum_j z_{jv}^t$ we relax problem (1) to

$$\min_{\boldsymbol{\beta}^t} \quad \sum_v \boldsymbol{\beta}_v^{t\,T} K^t \boldsymbol{\beta}_v^t - \frac{2}{\mathsf{M}_v} \mathbf{1}_{\mathsf{M}_v}^T K^{vt} \boldsymbol{\beta}_v^t \tag{2}$$

$$\text{s.t.} \quad \mathbf{0} \le \boldsymbol{\beta}^t \le \mathbf{1}, \quad \mathbf{1}^T \boldsymbol{\beta}_v^t = 1, \forall v, \quad \sum_v \beta_{nv}^t \le 1, \forall n$$

where $\boldsymbol{\beta}_v = (\beta_{1v}, \beta_{2v}, \cdots, \beta_{\mathsf{N}v})^T$, $K_{ij}^t = k(\boldsymbol{x}_i^t, \boldsymbol{x}_j^t)$, $K_{ij}^{vt} = k(\boldsymbol{x}_i^v, \boldsymbol{x}_j^t)$, and $\boldsymbol{x}_i^v$ is a training data point assigned to the $v$-th domain. Problem (2) is a convex quadratic programming problem which can be solved efficiently. We recover the indicator variables by $z_{nv^\star}^t = 1$ if $v^\star = \arg\max_v \beta_{nv}^t$.

After slicing the test datasets in this way, each slice/subset is also matched to a particular training domain in terms of the smallest distribution distance. Arguably, the adaptation between the matched pairs are easier than some arbitrary pairs or the original datasets.