[Reviews · NeurIPS 2013]

Submitted by Assigned_Reviewer_4

The goal of this work is to automatically discover latent domains in a training set, which is subsequently used in a domain adaptation framework to yield improved classification performance on a test set. The paper defines a function that measures the difference between two feature vectors over a specified kernel. The goal is to partition the data points into domains such that the function is maximized over the set of points across each pair of domains. The problem is formulated as an integer programming problem with two constraints: each point is assigned to exactly one domain and the distribution over class labels in each domain must match the input distribution over the entire point set. The problem is relaxed to a continuous optimization over a quadratic cost with linear constraints. Finally, the number of domains is found via cross validation.

The approach is evaluated over two datasets: static images of [2] and the IXMAS multi-view action dataset of [15]. A highlight is that improved performance is shown over the domain adaptation approach of [19].

Positives: The paper is well-written and as far as I'm aware the approach is novel (although I'm not an expert on domain adaptation). The performance gains over [19] is also appreciated.

Negatives: At this point I slightly lean towards reject. I have two main concerns that I would like to see addressed in the rebuttal that may convince me to change my score:

(i) The motivation for this paper is not clear to me. On line 77 the paper argues that "simply clustering images by their appearance is prone to reshaping datasets into per-category domains". First, what is the evidence for this claim? Second, how does the model formulation in Section 2 overcome this issue, i.e. how is not reshaping into per-category domains enforced?

(ii) Somewhat related, on line 154, why is the second constraint ("label prior constraint") needed? I'm curious what would the performance be without this constraint. In fact, a baseline where the data is partitioned using k-means clustering or unsupervised object discovery (e.g. Sivic et al ICCV '05) over the appearance vectors should be shown. Also, what is the performance when the dataset is randomly partitioned into equal sets?

Some additional comments:

+ Line 82, "maximally different in distribution from each other": This is mentioned throughout the paper. It would be good to clarify what this means. Distribution over what?

+ Line 166: Doesn't \beta_{mk} as formulated already live on a simplex?

+ Line 182: These seem to be different constraints than formulated before (starting on line 153), no?

+ Line 215: What is the justification/proof for this bound?

+ Line 289: Please provide more details on the use of the geodesic flow kernel [4]. Is this a reimplementation or was publicly available source code used?

+ Lines 313/340: Please provide some insights into the differences in performance. I want to better understand *why* the proposed approach is performing better. It would be good to show systematic failures of the baselines that the proposed approach overcomes.

+ Eq (1): M'_k => M_k'

+ This citation may be good to include as well:

Unbiased Look at Dataset Bias A. Torralba, A. Efros. IEEE Conference on Computer Vision and Pattern Recognition (CVPR), 2011.
Summary: The rebuttal addressed my concerns regarding the paper motivation and the label prior constraint. I lean slightly towards accept.

Submitted by Assigned_Reviewer_5

This paper presents a new technique for domain adaptation for computer vision datasets, which typically present multiple aspects, e.g. viewpoint, illumination, level of clutter, compression, etc. Instead of simply equating domains with datasets, which ends up mixing those aspects together, the proposed technique automatically partitions the set of all images over all datasets into domains. The partitioner is driven by two principles: making separate domains in feature space, and making them so that a good discriminative classifier can be trained on each of them (to classify the original classes, not to separate the domains). This technique is more likely to partition according to the underlying aspects rather than datasets.


Originality and significance:
There is only little work on automatically defining domains, and this paper proposes a good idea towards automatically discovering useful domains, that will support training better classifiers for the original problem.

On the negative side, I am not fully convinced of the proposed optimization method (section 2), as it's not clear how closely it solves the original problem (2)+(3). Moreover, the two proposed driving criteria are not well integrated yet: the maximal learnability criterion is only used as a 'wrapper around' the maximal separability criterion, in order to determine the number of domains. Essentially, it acts as a post-hoc validation score deciding how good is the domain partitioning learned for a given number of domains, but the partitioning itself is made based on the separability criterion alone. An integrated process would instead produce the partitioning that maximises some goal function including both criteria directly.

Despite these shortcomings, which might be due to the fact that this type of work is still at quite exploratory stages, I feel that the paper is a step in the right direction for the community and should be accepted.


Quality and clarity:
The paper is well written, but lacks figures to illustrate the concepts presented.


Experiments:
On the positive side, the experiments show a significant advantage in using the domains produced by the proposed method, over just using datasets as domains, and over the very recent domain discovery technique [19] (sec. 4.2).

The idea of including the test set in the 'reshaping' process is interesting, but not clearly presented (sec. 4.3). Also, this corresponds to an imputation setting (i.e. all test data is available at the same time), but this is not stated clearly.

On the negative side, the image descriptor used is very simple and outdated: just one bag-of-words of sparse SURF features for the entire image, and with just 800 codebook entries. This is really weak nowadays. I recommend the authors to use a spatial pyramid of bag-of-words, computed on _dense_ SURF features. Also, it is not clear what similarity measure is used to compare image descriptors, hopefully X^2 or an intersection kernel? The Euclidean distance is not suitable for comparing histograms. As a next step, a Fisher Vector representation could help further and finally place the image representation in this paper at the level of modern systems. The following papers might help the authors: Zhang IJCV 2006; Jurie CVPR 2005; Lazebnik CVPR 2006; Perronin ECCV 2010.
Summary: Overall, the paper presents interesting novel ideas on an important problem and achieve good results. The paper can be improved in several way, especially in terms of the image representation used.

Submitted by Assigned_Reviewer_6

This paper proposes a convex framework for splitting the dataset(s) into K subsets such that each subset has similar semantic class distributions and as distinct from each other as possible. Distinction, named as maximum distinctiveness, is achieved by maximizing the pairwise mean differences of subsets in the RKHS induced by a selected kernel. Each of these subsets are named as latent domains. Identification of K (number of latent domains) is achieved by maximizing average learnability (how well a classifier can be learned) within each latent domain.

Even though forcing the distribution of the classes within each domain to be as similar as possible prevents the clusters to be dominated by a single class each, it may also be a limiting assumption for latent domain discovery for certain tasks. For instance if we consider the poses as the latent domains for an animal classification task(e.g. horses and cows), then latent domains distribution within each class might not be similar. For instance, we observe both horses and cows in left-right standing pose, however horses are not pictured in a sitting pose often whereas the cows are.

Although the discovery of latent domains is not new, the idea of controlling class label distributions for better identification of latent domains within a (relaxed) convex framework is new. Identifying the number of latent domains through checking the quality of classification within each latent domain is also a notable practice which makes sure that the latent domain has enough number of samples for each class in order to better generalize and learn discrimination between classes.

In several places the concepts of dataset, domain and latent domain is not clear and can easily be confused. These concepts should be clearly defined, and preferably with some supporting examples. Particularly the experiments section 4.2. needs clarification. As far as I understand, the words dataset and domain is used interchangeably since S_i is both named as datasets and source domains. Nevertheless the experimental setting and the concept definition should be clarified in section 4.2. Additionally, max_k r(U_k,B) is not defined but used in eq.7.

The experimental validation appears to be adequate. The results have a reasonable improvement above the baselines. However, why the current selection of source and target datasets to report on is preferred is not clear. For instance leave one dataset out adaptation might be a more reasonable evaluation. The qualitative results are helpful to see what the algorithm visually achieves.

on line 074, the problem(learning latent domains) being stated as an unsupervised learning problem might be misleading since the methods use semantic class labels.
Summary: The discovery of latent domains via encouraging similar class distribution in each latent domain formulated in a (relaxed) convex framework is a notable technical contribution. However the some concepts in the paper and the experimental validation need to be clarified.
Author Feedback

Author rebuttal: We thank all reviewers for their comments. We are pleased with the comments on our approach being novel and original, with experiments that demonstrate its advantages.

== R4 ==

What is the motivation/evidence of per-category domains via simple clustering?

Our work models 3 interplaying forces: inter-category, intra-category and inter-(latent)domain variability. Visual appearance is often dominated by the first. As such, simple clustering by appearance can give rise to domains dominated by only one or a few visually similar categories. For example, one often observes and exploits so-called “discriminative clustering” when modeling image data, cf. Gomes, Krause, and Perona, NIPS 2010. Our work highlights and explicitly models the inter-domain variability, which is crucial to the domain adaptation setting where we expect such variability dominates. Our quantitative comparison to [20], which is a sophisticated appearance clustering approach, supports this claim.

Further analysis confirms too that simple clustering tends to align with class labels. We used the entropy of the class label distribution to measure how domains are aligned with class labels – small entropy implies strong alignment. For example, we found that the identified domains by K-means have an averaged entropy around 1.8 and 2.0 for (C, D,W->A) and (A, C->D,W) (of Table 1), while our method yields 2.3 for both.

How does formulation in Sec 2 avoid per-category domains?

The constraints Eq. (3, 5) force all identified domains to have the same label prior distribution as in the original datasets. Thus, having a single class or a small number of them in each domain violates the constraints.

Why the 2nd constraint on label prior?

In addition to preventing domains from being dominated by only a few categories, it simplifies the need to model “target shift” where prior distributions also change across domains (cf Zhang et al, ICML 2013). Removing it degrades performance: for (C, D, W->A) in Table 1, accuracy drops from 44.6% to 41.3%; for (A,C->D, W), from 42.6% to 39.3%.

How about a clustering or random partition baseline?

We confirm the finding by [20] that Kmeans clustering does not yield good domains: accuracies in Table 1 drop from 44.6% to 35.4% for (C,D, W->A), from 42.6% to 38.2% for (A,C->D, W). For randomly equal partitions, accuracies are reduced to 39.4% and 39.9%, respectively.

Specific comments:

L82: the distribution refers to the distribution over x, ie, the features.
L166: \beta_{mk} are vertices of the simplex before relaxation. We relax them to fill the interior.
L182: Correct. These constraints are derived from L53, to tighten relaxation and to lead to more tractable computation.
L215: the bound is used to ensure that we have at least one sample per category for cross-validation (to calculate A(K))
L289: we use the codes provided by [4] on web.
L313-340: Our insights highlight the very challenging issue of defining what a domain is. We believe our method works better as we define the domains as maximally distinctive from each other, yet being flexible so as not to commit to specific parametric forms for the distributions in latent domains, in stark contrast to the previous approach [20]. Given the complex visual data, the flexibility in modeling and the more principled approach of disentangling domains are key. As we see in the main text and in the results summarized above, Kmeans creates clusters dominated by a few classes, while random partitions create domains that are similar to each other (ie, equally underperforming) due to the lack of distinctiveness among them.

== R5 ===

How closely is original problem (2)+(3) solved?

We relax the NP-hard problem to continuous quadratic optimization for its scalability. The less scalable SDP relaxation would in theory give tighter bounds on optimality gap.

Tighter integration of two proposed criteria?

That could be interesting to pursue. For now, maximal learnability serves as the model selection criteria, regularizing the process of identifying latent domains. This is conceptually analogous to using BIC (or AIC) to select models instead of being added to objective functions as regularizers.

Experiments show significant advantage…[but] how about trying stronger features/kernels?

Thanks for the tips. We will explore those features. The current choices of SURF and shape-flow features are to ensure the fairest possible comparison to existing work [2,20,31]. We believe the choice of features is likely orthogonal to the choice of the latent domain discovery algorithm. We expect additional features to boost both our method and the baselines.

== R6 ==

Is the assumption/class distribution constraint limiting?

R6’s example highlights the need to utilize prior knowledge on specific latent factors (such as pose). Our current framework, while being general, nevertheless can leverage such knowledge, e.g, by constraining a subset of \beta_{mk} to be zero, thus, excluding them from participating in the constraint that enforces distribution match. This will be interesting future work. (Note, despite the generality, on two challenging datasets, the framework yields significant improvement in accuracy over other methods.)

Clarify in Sec 4.2
Thank you, we will distinguish the use of “domain” and “dataset”.

How source/targets selected to report on? How about leave one out?

We do report leave-one-dataset-out adaptation (Table 1 and 2: C, D, W→ A). We chose this split to avoid adapting between D and W, which are incidentally similar based on prior research [2,4], eclipsing the need to identify truly distinctive domains.

Other comments:

r(u_k, B) is the previously defined r(A, B), with A being replaced by u_k.
L74: Our algorithm does not know the true domain labels, thus “unsupervised”. Semantic class labels are used as side information but themselves are not domain labels.